# Aquatic Toxicity Effects and Risk Assessment of ‘Form Specific’ Product-Released Engineered Nanomaterials

**DOI:** 10.3390/ijms222212468

**Published:** 2021-11-18

**Authors:** Raisibe Florence Lehutso, James Wesley-Smith, Melusi Thwala

**Affiliations:** 1Water Centre, Council for Scientific and Industrial Research, Pretoria 0001, South Africa; flehutso@csir.co.za; 2Department of Chemical Sciences, University of Johannesburg, Johannesburg 2028, South Africa; 3Electron Microscope Unit, Sefako Makgatho Health Sciences University, Pretoria 0001, South Africa; jameswesleysmith@icloud.com; 4Centre for Environmental Management, University of the Free State, Bloemfontein 9031, South Africa

**Keywords:** nanotoxicity, engineered nanomaterials, nano-enabled products, risk assessment

## Abstract

The study investigated the toxicity effects of ‘form specific’ engineered nanomaterials (ENMs) and ions released from nano-enabled products (NEPs), namely sunscreens, sanitisers, body creams and socks on *Pseudokirchneriella subcapitata*, *Spirodela polyrhiza*, and *Daphnia magna*. Additionally, risk estimation emanating from the exposures was undertaken. The ENMs and the ions released from the products both contributed to the effects to varying extents, with neither being a uniform principal toxicity agent across the exposures; however, the effects were either synergistic or antagonistic. *D. magna* and *S. polyrhiza* were the most sensitive and least sensitive test organisms, respectively. The most toxic effects were from ENMs and ions released from sanitisers and sunscreens, whereas body creams and sock counterparts caused negligible effects. The internalisation of the ENMs from the sunscreens could not be established; only adsorption on the biota was evident. It was established that ENMs and ions released from products pose no imminent risk to ecosystems; instead, small to significant adverse effects are expected in the worst-case exposure scenario. The study demonstrates that while ENMs from products may not be considered to pose an imminent risk, increasing nanotechnology commercialization may increase their environmental exposure and risk potential; therefore, priority exposure cases need to be examined.

## 1. Introduction

Engineered nanomaterials (ENMs) can be released throughout the life cycle of nano-enabled products (NEPs) into the environment, and water resources are among the most common recipients [1]. The highest release of ENMs from NEPs (5–95%) has been estimated to occur during the product use stage [2,3]. Several studies confirmed the release of ENMs from various commercial NEPs, for example, textiles [4,5,6,7,8,9,10,11], paints [12,13,14], beauty products [15], and sunscreens [16,17,18,19]. The physicochemical properties of the product-released ENMs (PR-ENMs) can vary widely depending on the type and life cycle of the NEP [20]. Studies have shown that PR-ENMs can retain the characteristics they exhibited when incorporated into NEPs, form aggregates, remain embedded within the NEPs’ matrix, or undergo particle size change (reduction or increase) [21,22]. In general, studies show that the PR-ENMs sizes are found in the nano (<100 nm) and bulk (>100 to 385 nm) ranges and the amounts released are in the range of 0.01 to 35% (*w/w*) [23]. The confirmed release by several studies generally illustrates that NEPs are potential sources of water resource nano-pollution at different life cycle stages.

Studies investigating PR-ENMs in aquatic environments indicate that their presence in natural water environments can be predominantly linked to anthropogenic activities such as the release from nano-enabled sunscreens applied by swimmers or effluent discharges from ENMs and NEPs manufacturing industries [24,25,26,27,28,29,30,31,32,33,34,35,36]. The environmental exposure of PR-ENMs and subsequent bioavailability to organisms raises concerns due to limited information on hazard exposure potential and the uncertainty related to their risk. It is under this context that ENMs, PR-ENMs in this case are a class of emerging contaminants (nano-pollution) [37,38,39].

The uncertainty surrounding the environmental risk posed by PR-ENMs requires advancement of hazard and exposure assessment data. Regarding ecotoxicological effects, only a few studies have examined the effects of silver nanoparticles (nAg) from commercial textiles [40,41], titanium dioxide nanoparticles (nTiO_2_) [42,43,44,45] and zinc oxide nanoparticles (nZnO) [19,45,46,47,48] from sunscreens, nAg [49,50] from washing machine and cleaning products, as well as copper oxide nanoparticles (Cu_2_O) [13,51] and nAg [52] from paint, on various aquatic organisms. Available studies show that PR-ENMs induce toxicological responses ranging from negligible effects on enzyme activity, altered expression of antioxidant genes, abnormal development, growth inhibition, and mortality. However, results of studies on the effects of PR-ENMs are difficult to compare due to differences in experimental conditions and product types [22].

For example, the assessment of the toxic effects of sunscreen’s PR-nTiO_2_ has been widely diverse on organisms *Cyanobacterium* [43], *Prochlorococcus* [43], natural marine communities [43], *Paracentrotus lividus* [42], *Thalassiosira pseudonana* [44] and zebrafish embryos [45]. Furthermore, the methods used to obtain PR-ENMs also vary. In the case of sunscreens, PR-ENMs were generated either through extraction using organic solvents or obtained by simulating real-world release conditions [42,43,44,45]. For textiles, PR-ENMs were obtained by simulating washing conditions using media of different water chemistries. Hence, despite the accumulating evidence on the toxicity of PR-ENMs, the ability to compare the results between studies remains poor, thus hampering efforts to strengthen risk determination of PR-ENMs in aquatic systems [22,53].

To overcome experimental challenges, in silico approaches have been applied, which to date have largely concluded that PR-ENMs pose a low risk to aquatic organisms [54,55,56,57,58]. In sharp contrast, Musee [59] argued that modelling studies rely on assumptions that may not provide a true reflection of real environmental scenarios. For example, in those highlighted in silico studies, risks were estimated using predicted environmental concentrations (PECs), which are often lower than measured environmental concentrations (MECs) and therefore result in low-risk characterisation [59]. The PECs of nAg and nTi in aquatic resources were reported at 0.014–2.2 µg/L [54,60,61] and 0.7−16 µg/L [2,54,56,61], while the MECs were quantified at 0.03–19.7 µg/L and 0.67–150 µg/L, respectively.

With increasing advances in nanotechnology, the production and use of ENMs in commercial products increases the probability for environmental release, leading to MECs increasing, which may end up reaching levels that induce aquatic toxic effects [62,63]. Consequently, current in silico risk predictions may underestimate the risk of PR-ENMs as environmental exposure concentrations increase.

A further shortcoming in most studies (in silico and laboratory) to date has been the use of pristine ENMs’ ecotoxicological data and not PR-ENMs’, although the two counterparts are known to exhibit different physicochemical and hazardous properties [64,65]. For example, modification of the surface of ENMs with aluminium and silicon-based coatings in preparation for incorporation into NEPs alters their physicochemical characteristics [66,67,68]. Their properties are further altered by the external stressors encountered by NEPs during their use and by environmental conditions [20]. This is illustrated by the effect of washing and weathering solutions that induce ENMs’ transformation in nano-enhanced textiles and paints [5,69]. Similarly, once released into aquatic environments, ENMs interact with environmental factors and undergo chemical, physical, or biological transformation [22,70,71], which can change the mobility and bioavailability of ENMs [71]. It is such factors that can influence the effects of PR-ENMs from pristine-ENMs and consequently the risk; hence, the use of pristine-ENMs ecotoxicological data to predict the risk of PR-ENMs may lead to incorrect risk estimation.

The current study investigated the toxicity effects and risks of ‘form-specific’ PR-ENMs and associated ions. In this case, PR-ENMs refer to ENMs released from NEPs, whereas the associated ions refer to ions also isolated from PR-ENMs. The ‘form-specific’ ENMs and associated ions were released from various NEPs classified as having a medium to high potential for release into aquatic environments [72]. Furthermore, the study assessed the interactions between sunscreen PR-ENMs (as a case study) and aquatic biota (*Pseudokirchneriella subcapitata*, *Spirodela polyrhiza*, and *Daphnia magna*) by ultrastructural histology examination following exposure. The selected species were considered suitable for the purpose of the study as they belong in different aquatic trophic levels and are hence representative of various ecological groupings. Furthermore, they are well-established sensitive standard test models for investigating aquatic stress [73,74], including assessment of nano-pollution [75,76]. The PR-ENMs samples were not purified to remove other ingredients that may contribute to or influence the toxicity of PR-ENMs, especially sunscreen, sanitisers, and body cream products; this was due to the highly complex exercise of leaching and characterising the bioavailability and toxicity contribution of each NEPs’ ingredients. It is for this reason, the study paid particular attention to investigating the toxic effects of the primary inorganic filters and antimicrobial agent, which are nZnO, nTiO_2_ and nAg [77]. The isolation and contribution of each product ingredients to the environmental behaviour and effects of PR-ENMs remains a major scientific challenge in addressing the risks of ENMs [78,79]; however, toxic effects of NEPs have closely been linked to the presence of ENMs despite the potential contribution of other ingredients [19,78,80].

## 2. Results and Discussion

The interaction and effects of specific ionic and particulate forms released from six consumer products as well as the respective risks were assessed on *D. magna*, *P. subcapitata*, and *S. polyrhiza.* The study investigated the acute toxicity effects of PR-ENMs and ions at 24 and 48 h for *D. magna* and 72 h for *P. subcapitata and S. polyrhiza* as prescribed by the Organization for Economic Cooperation and Development (OECD) and the International Organization for Standardization (ISO) test guidelines [81,82,83]. The findings pertaining the toxicity and risk are provided in Section 2.1 and Section 2.2, whereas Section 2.3 pertains interactions with biota.

### 2.1. Toxicity Effects of PR-ENMs and Ions (% v/v)

#### 2.1.1. Dose–Response

##### *P.* *subcapitata*

The PR-ENMs induced varying toxic effects on *P. subcapitata* in descending order: SAN1 > SUN1 > SK1 > SUN2 > SUN3 > CA1 (Figure 1A). The SUN1–3 PR-ENMs obtained under light and dark conditions induced comparable effects, indicating no influence by light variation. For CA1, higher toxic effects were observed from PR-ENMs obtained under light conditions. The ions released from SUN1, CA1, SAN1 and SK1 also induced varying toxic effects on *P. subcapitata* (SUN1 = SAN1 > SK1 > CA1). SUN1 and SAN1 ions were relatively more toxic, while CA1 ions induced negligible effects on *P. subcapitata* compared to the control. In fact, the growth inhibition by CA1 ions was less than 50% relative to the control and therefore not included in Figure 1A.

The released ions were found to have contributed to the overall PR-ENMs’ toxicity (Figure 1B). The effects of SUN1 ions (*p* = 0.006) and SK1 ions (*p* = 0.02) were significantly different from the overall PR-ENMs mixture, indicating that the ions induced either synergistic or antagonistic effects. SAN1 ions induced additive effects since the predicted and observed effects were comparable.

The particulate fraction also contributed to the toxic effects (Figure 1B). SUN1 particulates induced additive effects, while their SK1 and SAN1 counterparts induced synergistic or antagonistic effects (Figure 1B). Overall, based on the assumption that exposure to a mixture of ions and particulates coincides and exhibits a unique toxicological mode of action [84,85]; the findings suggested that both particulates and ions contributed to the induced toxic effects. The main component(s) responsible for the toxicity of pristine-ENMs and ions, alone or in combination, has been investigated in aquatic organisms, and contradictory results have been reported, although the common conclusion was that both ENMs and ions contribute to the induced effects [86,87,88].

##### *S.* *polyrhiza*

Growth of *S. polyrhiza* was inhibited by all PR-ENMs to varying degrees (Figure 2A). The overall toxicity trend in descending order was SUN1 > SAN1 > SK1 > SUN3 > SUN2 > CA1. The variation of illumination did not influence the toxicity of SUN1–3 PR-ENMs; however, in the case of CA1 PR-ENMs obtained under light conditions, toxicity was enhanced. The effects of PR-ENMs and associated ions obtained under light (*p* = 0.0001) and dark (*p* = 0.01) were found to be significantly different for SUN1, CA1 (*p* = 0.006) and SAN1 (*p* = 0.0082), while for SK1, the effects were comparable (*p* = 0.12). The ions released from NEPs induced either synergistic or antagonistic effects (Figure 2B). No definitive trend could be established between the toxicity potential of ions vs. PR-ENMs, as toxicity varied between products (based on the comparison of EC_50_s). For SUN1, the ions were more toxic (EC_50_: 1.4–1.9%) than their PR-ENMs counterparts (EC_50_: 9.5–9.7%). On average, this was also the case with SAN1, indicating that the ions were the primary toxicity agent. On the contrary, the CA1 and SK1 PR-ENMs were relatively more toxic than the ions, indicating that the ions were not the main toxicity agent. The particulates in the CA1, SAN1, and SK1 exposure media were found to exert either synergistic or antagonistic effects.

##### *D.* *magna*

All PR-ENMs immobilised *D. magna* to varying degrees after 24–48 h exposure (Figure 3A). After 24 h, the effects induced by SAN1, SUN1, SUN3 and SK1 PR-ENMs were similar, while CA1 PR-ENMs were the least toxic: SAN1 = SUN1 = SUN3 = SK1 > SUN2 > CA1. Generally, toxicity increased with the duration of exposure. After 48 h exposure, the toxicity trend in descending order was PR-ENMs SA1 = SUN1–3 = SK > CA1. The released ions were also toxic: their overall toxicity trend (after 24 h) was in descending order: SAN1 > SUN1 = SK1 > CA1. Similar to PR-ENMs, the toxic effects induced by the ions increased with the duration of exposure, and after 48 h, the toxicity trend of the released ions was similar to that of PR-ENMs. A comparison between PR-ENMs and ions showed that the effects of SUN1 (both light and dark cycles), SK1, and SAN1 PR-ENMs, as well as associated ions, were comparable after 24 h exposure. The PR-ENMs of CA1 (irrespective of illumination conditions) were more toxic than ions. After 48 h exposure, SUN1, SK1, and CA1 PR-ENMs (irrespective of illumination conditions) were more toxic than the released ions, while the toxic effects of SAN1 PR-ENMs and ions were comparable. Overall, the ions and PR-ENMs caused synergistic or antagonistic effects, indicating that the toxic effects on *D. magna* were due to a combination of both (Figure 3B,C). 

#### 2.1.2. Species Sensitivity

The three test organisms exhibited varying degrees of sensitivity to toxicants (PR-ENMs and ions). The order of sensitivity to SUN1–3 and SK1 PR-ENMs was *D. magna* > *P. subcapitata* > *S. polyrhiza* (Figure 4). For CA1 and SAN1 PR-ENMs, the trend was *D. magna* > *S. polyrhiza* > *P. subcapitata*, and *P. subcapitata* > *D. magna* > *S. polyrhiza*, respectively. The species sensitivity distributions (SSDs) of the released ions were mostly similar to the respective PR-ENMs, except for SAN1 (Appendix A). SAN1 ions followed a pattern similar to that of SUN1–3 and SK1 PR-ENMs. Since CA1 ions induced adverse effects below 50%, *P. subcapitata* was considered least sensitive. Overall, the results illustrated the varying extent of aquatic biota vulnerability to nanopollution, indicating that *D. magna* potentially faces elevated risk, followed by *P. subcapitata* and least of all *S. polyrhiza*.

### 2.2. Toxicity Assessment of PR-ENMs (Relative to Quantified Amounts)

#### 2.2.1. Toxicity Evaluation of Relative Contribution of Binary PR-ENMs

For exposure media containing binary PR-ENMs (SUN1, CA1, and SK1), a mixture analysis approach (probability theory) was adopted and used as a predictive tool to select the concentration of PR-ENMs to be used to calculate the EC/LC_50_ (µg/L). For this analysis, it was hypothesised that organisms are exposed to binary PR-ENMs simultaneously. The results indicate that the respective binary PR-ENMs obtained from SUN1, CA1, and SK1 contributed to the overall toxicity in the three organisms (Appendix A, Appendix A) and hence, herein, the individual PR-ENMs EC/LC_50_s (ug/L) were not determined. EC/LC5_0_ of SUN2–3 and SAN1 PR-ENMs, and SUN1, SAN1, SK1, and CA1 ions, were calculated using the exposure concentration of the target analyte quantified by inductively coupled plasma mass spectrometry [15].

The EC/LC_50_ of binary PR-ENMs have been determined elsewhere using the total concentration of the PR-ENMs of interest [19,47]. For example, the LC_50_ of binary sunscreen PR-ENMs (nTiO_2_ and nZnO) was calculated using the total concentration of Zn; the total concentration of Ti was neglected because nZnO was considered to be the primary toxic agent [19]. In the study of Schiavo et al. [47], interest was directed towards the effects of PR–nTiO_2_ and not PR–nZnO, and therefore EC_50_ was calculated using total Ti.

The different approaches used in calculating the EC/LC_50_ for binary PR-ENMs have serious implications for risk assessment outcomes for such NEPs. This, as EC/LC_50_ data, is used to determine PNECs, which, in turn, is used to estimate risk. As shown herein, the presence of binary PR-ENMs in exposure media plays a crucial role in the induced toxic effects. Therefore, the determination of EC/LC_50_ using a concentration of only one of the present ENMs in mixtures should be reconsidered. Recently, Bicherel et al. [89] proposed a method that factors the concentration of each constituent in the exposure media to estimate the toxicity of the chemical mixture. Due to the complexity associated with translating water-accommodated fraction loading rate results into PNECs in different compartments, the method is currently incapable of estimating the risk; the method is under continuous development.

#### 2.2.2. Half-Maximal Effective or -Lethal Concentration

The EC/LC_50_s of the PR-ENMs and ions are provided in Table 1. For all test organisms, the SAN1 PR-ENMs were the most toxic; the toxicity descending order was: SAN1 > SUN2 > SUN3 for *P. subcapitata* and *D. magna*, while for *S. polyrhiza* the toxicity descending order wasSAN1 > SUN3 > SUN2 for and, respectively. The EC/LC_50_ of SAN1, SUN1, CA1, and SK1 ions varied. Similar to PR-ENMs, SAN1 ions induced the highest toxic effects in all test organisms. In descending order, the toxicity of the ions was: SAN1 > SUN1 > SK1 > CA1 for *P. subcapitata* and SAN1 > CA1 > SUN1 > SK1 for *S. polyrhiza* and *D. magna*.

The EC/LC_50_s of the PR-ENMs in previous studies are mostly not reported; only a few studies reported the EC/LC_50_ on aquatic organisms. In the algae taxonomic group, 24 h EC_50_ (*Dunaliella tertiolecta*) was reported at 25–34 µg/L after exposure to PR-nTiO_2,_ while 72 h exposure did not cause an adverse effect; instead, a biostimulating effect was observed [47]. For *Phaeodactylum tricornutum*, the 72 h EC_50_ was greater than 100 and 9.9 µg/L after exposure to sunscreen PR-ENMs (nTiO_2_+nZnO) [90]

For crustaceans, the marine copepod *Tigriopus japonicus*, 96 h LC_50_ of 43–>5000 (mg/L) and 1.2–82.5 (mg Zn/L) were recorded after exposure to sunscreen PR–ZnO [19]. Using *Crataegus*
*orientalis*, the 96 h LC_50_ was found to be higher than the maximum tested concentration (100 µg/L) and 8.7 µg/L under different salinity conditions [90]. For fish, the 24 h LC_50_ of 140 µg/L and 260 µg/L were recorded after exposure to socks PR–nAg [41]. Although the reported EC/LC_50_ were determined in organisms that differed from the current study, the EC/LC_50_s determined in the current study were consistent with the values reported for the respective organism taxon group. In general, EC/LC_50_ data for aquatic organisms for PR-ENMs are minimal and continue to hamper efforts to establish PR-ENMs risk assessment efforts, which are currently filled with significant data gaps [91].

#### 2.2.3. Predicted No Effects Concentrations of PR-ENMs

The PNECs of the PR-ENMs and the ions were successfully determined (Figure 5). The PNECs determined for both the PR-ENMs, and the ions differed between the NEPs. The PNECs of SAN1 PR-ENMs were determined to be 1–<5 ng/L for *P. subcapitata* and *D. magna*, while it was approximately 100 ng/L for *S. polyrhiza*. For SUN2–3 PR-ENMs, the PNECs ranges were 80–200, 100–200, and 5–10 ng/L for *P. subcapitata*, *S. polyrhiza*, and *D. magna*, respectively. The PNECs of all NEPs ions (SAN1, CA1, SK1, and SUN1) ranged between 9–<700 µg/L, 32–2046 µg/L, and 1–240 ng/L for *P. subcapitata*, *S. polyrhiza*, and *D. magna*, respectively.

The range of PNECs for PR-ENMs (nTiO_2_ and nAg) found in the current study was comparable to reports from modelling studies. For example, PNECs of nTiO_2_ released from various NEPs into aquatic environments (algae and daphnia) were estimated to be less than 1000 ng/L [2]. Elsewhere [61], the PNECs of cosmetics PR-nAg and PR-nTiO_2_ were determined to be 6.69 ng/L *(Ceriodaphnia dubia*), 100–3000 ng/L (*D. magna*) and 900 ng/L (*P. subcapitata*). To the best of our knowledge, PNECs of PR-ENMs (both experimental and modelled data) in aquatic environments specific to *S. polyrhiza* and higher aquatic plants have not been reported before.

#### 2.2.4. Risk Characterisation of PR-ENMs

The risk assessment of PR-ENMs towards the three test organisms indicates that no imminent risk (RQ < 1) is expected in all cases. For all test organisms, the risk estimation indicated that SAN1 PR-nAg posed potential adverse effects (RQ > 100). SUN2’ PR-nTiO_2_ posed small to potential adverse effects (RQ = 1–10) on *P. subcapitata* and *S. polyrhiza.* For *D. magna* at 24 h exposure, SUN2–3 PR-nTiO_2_ were estimated to pose significant adverse effects (RQ = 10–100). The risk of toxicity for *D. magna* increased with prolonged exposure; at 48 h exposure, SUN2–3 PR–nTiO_2_ are expected to pose potential adverse effects (RQ > 100).

Similar to PR-ENMs, no imminent risk (RQ < 1) is expected in aquatic environments (all three test organisms) from the SAN1, SUN1, CA1 and SK1 ions. Significant adverse effects (RQ = 10–100) are expected from the SAN1 and SUN1 ions on all test organisms; SAN1 ions at 24 and 48 h exposure are expected to cause potential adverse effects (RQ > 100) on *D. magna*. The CA1 and SK1 ions are expected to pose negligible to potential adverse effects on *D. magna*, *P. subcapitata*, and *S. polyrhiza*, respectively. At 24 and 48 h exposures for *D. magna*, CA1 ions are expected to cause small to potential adverse effects, while SK1 ions are expected to pose significant adverse effects.

The RQ obtained in the current study agrees well with modelling exercises; in all cases, imminent risk is not expected. Modelling studies have so far predicted mainly small to potential adverse effects (RQ = 1–10) from cosmetics’ PR-nAg and PR-nTiO_2_ in aquatic ecosystems of Johannesburg, South Africa [61] and NEPs PR-nAg in freshwater of Europe and Switzerland [56]. However, in the broader spectrum, the risk profile observed herein slightly differs from the findings of most modelled studies, in that PR-ENMs risks are considered low risk in aquatic environments [57,92]. The difference between the findings of the modelling studies and the current study can be attributed to the use of PECs and MECs, respectively. The MECs used herein were higher than the reported PECs, in some cases by a factor of 1000. Higher MECs than PECs of nAg, nTiO_2_, and nZnO in the environment have also been reported elsewhere [93]. Thus, using MECs instead of PECs in the RQ calculation provided higher RQ values [59].

The risk profile obtained from MECs and PECs can be viewed as, respectively, narrating a worst- and least-case scenario, for instance, at the point of discharge and later in the receiving natural environments. Meaning that the risk may be significant at the discharge points (or immediate surroundings) and decreases as the PR-ENMs travel further away from the release point and upon dilution. This statement is supported by modelling studies, in which the RQ values obtained for wastewater treatment plant effluent were between 1–10 for nTiO_2_, >10 <100 for nAg, and between 1 > 10 for nZnO [56]. Additionally, a recent experimental study found that an environmentally relevant concentration (25 µg/L–very dilute) of NEPs PR–nTiO_2_ poses a negligible risk to the cyanobacterium *Prochlorococcus* and natural marine communities [43]. These findings suggest that the ‘one size fits all’ conclusion that PR-ENMs are of little concern in the aquatic environment should be treated with caution; instead, risk assessment and conclusions should be associated with a specific exposure scenario. Current data on toxicity risk characterisation are limited to modelling, and little to none are founded on experimental analysis. More experimental data are required for a conclusive risk assessment of PR-ENMs, which can then be used in conjunction with modelling studies to produce reliable and robust risk assessments of NEPs.

### 2.3. Sunscreen PR-ENMs Interaction with Biota

#### 2.3.1. Interaction with *P. subcapitata*

The SUN1 PR-ENMs induced slight morphological deformations (normal to crescent shape) compared to controls (Figure 6A,B). Control samples of *P. subcapitata* were in a healthy state with organelles such as mitochondria, chloroplasts, starch grains, cell walls, plasma membranes, vacuoles, nuclear envelope, and nuclei visibly intact (Figure 6A). Despite slight morphological deformation, no damage to the cell wall or plasma membrane was observed (Figure 6B). No internalisation of SUN1 PR-ENMs was observed, but PR-ENMs aggregates were observed bound externally to the cell walls. SUN2 PR-ENMs did not cause structural cell damage, but promoted increased lipid body production (Figure 6B). Similar to SUN2 PR-ENMs, SUN3 PR-ENMs did not cause significant changes in the morphological integrity of *P. subcapitata*, although an apparent increase in starch grain production was observed (Figure 6C). Excessive starch grains and lipid bodies production occurs when algae and higher plants experience stress, a phenomenon not unique to ENMs. The increase in starch grains and the production of larger grains has been reported in algae and higher aquatic plants after exposure to various ENMs [94,95]. Several authors reported increased lipid bodies and starch grains mainly occur under oxidative stress conditions [96,97,98], a toxicity mechanism commonly reported in nanotoxicity [99,100].

Similar to SUN1, SUN2–3 PR-ENMs were also not internalised but were adsorbed on the cell walls. The dark and bright spots in Figure 6 are PR-ENMs; images were captured under bright and dark fields. Internalisation of ENMs has been suggested to likely occur in scenarios where ENMs are smaller than the cell pore size [101,102]. SUN1–3 PR-ENM sizes (Table 1) were similar to or slightly greater than the diameter of the algal cell pores estimated to be 5–20 nm [101]. The lack of ENMs internalisation by *P. subcapitata* has previously been reported for nCeO_2_ and nSiO_2_ [102,103]. In the absence of ENMs internalisation and release of ions in the exposure media and no light restriction, the observed effects were probably oxidative stress-induced due to ENMs aggregates adsorbed to cell surfaces [103].

#### 2.3.2. Interaction with *S. polyrhiza*

The *S. polyrhiza* control sample did not have or was not associated with PR-ENMs as observed from the analysis with SEM-EDX (Appendix A). SUN1 PR-ENMs (ZnO) were adsorbed on the roots and fronds of *S. polyrhiza* after exposure to SUN1 PR-ENMs, whereas PR-TiO_2_ were not detected (Appendix A). Although SUN1 PR-TiO_2_ were not detected, SUN2 PR-TiO_2_ were adsorbed on roots and fronds (Appendix A), while SUN3 PR-TiO_2_ were only detected on the roots (Appendix A).

Regardless of the type of PR-ENMs, the adsorption of PR-ENMs on *S. polyrhiza* was minimally detected. Nonetheless, the PR-ENMs were mainly observed on the root cap and hairs than on the fronds. This observation was not surprising since the roots were fully submerged in the exposure media and reached the bottom of the test container while the fronds floated on top of the exposure media. The uptake of ENMs by roots has been identified as a common entry path in plants [104,105], after which they translocate through vascular tissues to other parts of plants (i.e., fronds) [106]. The adsorption of pristine-ENMs on roots, fronds, and leaves of higher aquatic plants has been reported [94,95,107,108,109].

Although plants’ interaction with ENMs can induce adverse effects in plants, they can also induce growth stimulation for metal-tolerant plants [110]. Positive, adverse or insignificant effects of pristine-ENMs have been reported [109,111]. The adverse effects caused by ENMs include growth inhibition (roots and fronds), structural deformation, cell wall damage, plasmolysis, degraded endoplasmic reticulum, accumulation of starch grains, and reduced thylakoids [94,96,112,113,114,115,116,117]. Herein, in addition to reduced growth (Section 2.1), structural deformation, starch grains, and lipid production were some of the adverse effects observed (Figure 7).

Deposits suspected of being PR-ENMs adsorbed on the root epidermis (Figure 7B,C), and the cell wall (chain-like structure) and vacuoles of the fronds were observed, indicating a possibility of PR-ENMs internalisation (Figure 7F). In the absence of elemental analysis, the suspected PR-ENMs observed in the cross-section were not confirmed; therefore, it is impossible to accurately conclude the identity of the deposits. Similar observations of the deposition of ZnO-like particles on the cellular matrix in a chain-like shape, and the presence of ZnO nanoparticles aggregates around the cell membrane has been reported elsewhere [113,118].

#### 2.3.3. Interaction with *D. magna*

SUN1–3 PR-ENMs caused significant morphological damage to *D. magna* compared to the control (Figure 8). Elemental mapping showed higher amounts of SUN1 PR-nZnO (red colour) compared to SUN1 PR-nTiO_2_ (green colour) adsorbed on the surface of *D. magna* (Figure 8B). Additionally, cross-sectional samples found that Zn particles (probably including nZnO aggregates) were attached to the carapace (Figure 8B). Similarly, SUN2 PR-nTiO_2_ adsorbed on test organisms and were observed on the theropods and abdominal setae (Figure 8C). SUN3 PR-nTiO_2_ were also found on *D. magna* (Figure 8D), but due to structural deformation, the organelle on which the PR-nTiO_2_ particles were located on could not be positively identified.

For microscopic analysis, the amount of PR-ENMs within specimens was often below the resolution limit of the EDS system, which made detection and especially mapping difficult. Conducting spot analysis and reducing the area on the map improved detection. The reduced detection could be due to the loss of PR-ENMs, which could have occurred during the preparation of the biota sample. Elsewhere [119], similar analytical challenges were experienced when assessing ENMs-biota interactions, pointing to the limited capability of conventional analytical tools.

## 3. Materials and Methods

### 3.1. Preparation of PR-ENMs’ Samples

The materials investigated for toxicity effects and risk were previously characterised elsewhere [15]. Briefly, PR-ENMs were obtained from commercial sunscreens (SUN), body creams (CA), sanitisers (SAN) and socks (SK) (Table 2). The NEPs were sourced from South African retailers and were previously confirmed to be nano-formulated [120]. The procedures used to release ENMs from NEPs, isolate the released ions (from SUN1, CA1, SK1 and SAN1); the physicochemical properties of the PR-ENMs are extensively described elsewhere [15]. While PR-ENMs refer to ENMs released from NEPs, for toxicity investigations, it should be noted that the PR-ENMs media is ‘as released’, meaning it contained all ingredients released from the NEPs. Table 2 provides brief details of the physicochemical properties of the PR-ENMs and ions examined herein for their potential toxic effect and risk.

### 3.2. Toxicity Effects Assessment

#### 3.2.1. Dose–Response

*P. subcapitata*, *D. magna*, and *S. polyrhiza* were used as models to test the toxicity effects of PR-ENMs and ions.

In all toxicity investigations, the effects of the PR-ENMs were compared to the control populations, which were not exposed to ENMs. The tests were considered to have been successful only when the control samples met the standard validity criteria. For *P. subcapitata*, growth by a factor of 16 times after 72 h exposure compared to time 0 was required; for *S. polyrhiza*, the mean growth of the first fronds in control (octuplicate) after 72 h should average a minimum of 10 mm^2^*;* for *D. magna*, the percentage of immobilisation of the control sample should be less than or equal to 10%; the results reported herein met the respective test validity criteria.

Due to the continuous transformation of PR-ENMs, toxicity investigations were initiated immediately after the release procedure ended. In most cases, the organism’s toxicity was investigated in (i) PR-ENMs and associated ions within 30–60 min of release.

The test organisms were exposed to 5 concentrations (10-fold dilution, 0.01 to 100%) in triplicate in all instances. A 10-fold dilution series was selected based on preliminary empirical toxicity investigations, which showed that a 2-fold dilution series induced toxic effects at 1/8th concentration for most PR-ENMs.

##### *P.* *subcapitata*

The toxic effects of PR-ENMs and ions were investigated on *P. subcapitata* using the 72 h acute test protocol of Slabbert et al. [121]. Specimens of *P. subcapitata* were subcultured from a continuous in-house stock. To ensure that the cultures were in an exponential growth phase when used in toxicity tests, 10 mL of the stock cultures were regrown in 250 mL of 10% BG-11 medium at 25 °C and shaken on an orbital shaker (Labotech, South Africa) at 100 rpm for six days under light (6000 lux). On the sixth day, the experimental cultures (inocula) were prepared by transferring 1 mL × 24 of the regrown culture to Eppendorf tubes (Inqaba, South Africa), centrifuged at 10,000 rpm for 10 min, and the supernatants were discarded. Algal cells were resuspended in 100 µL BG-11 medium, vortexed for 30 s, centrifuged at 10,000 rpm for 8 min, and the supernatants were discarded. The procedure was repeated three times. The total volume of algae culture required for the entire test and the density of the algae were determined following the guidelines of Rodrigues et al. [122]. The 72 h toxicity tests were performed at 25 °C, shaken at 100 rpm, under 6000 lux light. The effects of PR-ENMs or ions were compared to the control [sterilised Milli-Q water (18 MΩ.cm)]. Cell density (in the form of optical density) was measured at times 0 and 72 h using a microplate reader (FLUOstar Omega, BMG, Germany) at 684 nm. For this measurement, 100 µL of the sample test were transferred to a 96-well plate. The 72 h EC_50_ (growth inhibition) was calculated using the ALGALTOXKIT software.

##### *D.* *magna*

The toxic effects of PR-ENMs and ions on *D. magna* were investigated following the OECD acute (48 h) test method [82]. Briefly, *D. magna* cultures used in the investigation were housed in an aerated water tank (2.5 L) maintained at 20–22 °C at a 16: 8 h light: dark cycle. For PR-ENMs and ions exposure, five individual *D. magna* neonates (less than 24 h old) were transferred from the culture tanks to a glass beaker containing 20 mL of exposure media (PR-ENMs or ions and control (culturing water)) and incubated under conditions similar to culture tanks. Adverse effects, specifically immobilisation of *D. magna*, were recorded at 24 and 48 h, and the LC_50_ was calculated using the MBT Daphnia Regtox software.

##### *S.* *polyrhiza*

For *S. polyrhiza*, the toxicity effects of PR-ENMs and ions were investigated following the MicroBio Tests Inc, Belgium adapted ISO 20227 method [81]. Briefly, dormant vegetative turions of *S. polyrhiza* were incubated at 25 °C for 72 h in 30 mL Steinberg media under continuous illumination (6000 lux). After 72 h, germinated turions, with small fonds, were transferred to 48 multiwell test plates using a spatula, each well contained 1 mL of sample (PR-ENMs or ions) and control (Steinberg media); investigations were performed in octuplicate. The exposure test was performed under conditions similar to those used to germinate the turions. A digital image of the multiwell plate containing the germinated turions in the samples was captured at T_0_ (beginning of exposure) and again after T_72_ (end of exposure). The frond areas for both images captured at T_0_ and T_72_ were measured using ImageJ software (NIH, Wisconsin, United State of America), and the 72 h growth inhibition EC_50_ was calculated using the *Spirodela* Regtox software.

#### 3.2.2. Evaluation of Relative Toxicity Contributions between Mixtures

Relative toxicity contribution of mixtures (PR-ENMs and ions) from SUN1, CA1, SAN1 and SK1 were examined using a response addition model (RA) [84,85]. The contributions were determined using the RA model; defined as E_(total)_ = 1 − [(1 − E_(ion)_)(1 − E_(particle)_)], where E_(total)_ and E_(ion)_ represent the toxicity caused by the suspensions of PR-ENMs and their corresponding released ions (scaled from 0 to 1) [84,85]. E_(total_) and E_(ion)_ were obtained from experimental toxicity data, while E_(particle)_ was calculated directly. Herein, E_(particle)_ represented both the nano-sized and bulk-sized particles released from the NEPs and are collectively classified as particulates or particulate fractions.

Since SUN1, CA1 and SK1’s NEPs of SUN1, CA1, and SK1 released binary PR-ENMs [15], the principal toxicant between the two or combined toxicity contribution of the binary PR-ENMs was determined using a predictive model described elsewhere [123,124]. Briefly, the predictive model is based on probability theory, defined as *P*(E) = *P*_(x)_ + *P*_(y)_ − (*P*_x_**P*_y_/100), where *P*_x_ and *P*_y_ are the inhibition induced by chemical x and y, respectively [123,124]. The procedure compares the effects measured in the experiment, referred to as the observed effect, *P*(O), with the theoretically expected/predicted effect *P*(E) [123,124]. The results were considered to have a synergistic or antagonistic effect when the observed effects [*P*(O)] and the theoretical effects [*P*(E] were found to be significantly different (*p* < 0.05). Insignificant differences (*p* > 0.05) between *P*(O) and *P*(E) were considered additive effects.

### 3.3. Species Sensitivity Distribution

Species sensitivity between *P. subcapitata*, *D. magna*, and *S. polyrhiza* after exposure to PR-ENMs and ions was determined using the Environmental Protection Agency (EPA) SSD generator [125]. The obtained toxicity data of the PR-ENMs and ions per NEP were individually exported to the SSD generator and the SSD software was processed according to the EPA SSD generator instruction [125]. The SSD was plotted as the cumulative probability (*y*-axis), calculated as the fraction of species affected at a specific concentration. The *x*-axis was the EC/LC_50_ determined in the present study at 72 h exposure for *P. subcapitata* and *S. polyrhiza* and 48 h for *D. magna.*

### 3.4. Risk Characterisation

Predicted no effect concentrations (PNECs) of the PR-ENMs and ions were determined according to the Technical Guidance Document (TGD) [126]. The PNECs were determined as the ratio of EC/LC_50_ to the assessment factor (AF). For this exercise the assessment factor of 1000 was applied since toxicity assessments were examined under acute conditions [126].

The risk of the PR-ENMs and ions was determined using risk quotients (RQ) method [127]. The RQ was determined as the ratio of MEC to PNECs. The total concentrations of PR-ENMs and ions (Table 2) were used as MECs. For the interpretation of the data on the potential risk posed, RQ < 1 implied no significant risk, RQ 1–10 implied small adverse effects, RQ 10–10 implied significant adverse effects, and RQ > 100 indicated potential adverse effects.

### 3.5. PR-ENMs Interaction with Test Organisms

The interaction between test organisms and PR-ENMs was assessed after exposure to SUN1–3 PR-ENMs. For *P. subcapitata*, interaction assessments were performed in algal cells exposed to 10% (*v*/*v*) PR-ENMs, because 100% (*v*/*v*) exposure induced 100% growth inhibition. For *S. polyrhiza* and *D. magna*, interaction evaluations were conducted on test organisms exposed to 100% (*v*/*v*) PR-ENMs.

#### Electron Microscopy Analysis Sample Preparation

Biota samples were collected and chemically fixed after exposure to PR-ENMs and controls. Briefly, 10 mL of *P. subcapitata* algal cells were collected and centrifuged at 10,000 rpm for 30 min. The supernatants were discarded and *P. subcapitata* pellets were fixed with 2.5% glutaraldehyde (Merck, South Africa). For *S. polyrhiza* and *D. magna*, the organisms were collected from the test containers using a spatula and a Pasteur pipette, respectively. Similarly, a sufficient volume of the fixative was added to cover the whole organism. In all cases, the chemically fixed organisms were kept at 4 °C until processed for electron microscopy analysis.

Fixed samples were pre-treated following a modified standard biological sample preparation protocol for electron microscopy analysis [128]. For scanning electron microscopy (SEM) coupled with energy-dispersive X-ray spectroscopy (SEM-EDX, Zeiss Supra 55VP, Munich, Germany), fixed biota samples were gradient dehydrated with ethanol (Merck, Johannesburg, South Africa). The algal suspensions were filtered using Millipore filters, and the ethanol was replaced with hexamethyldisilazane (Merck, Johannesburg, South Africa) in a stepwise manner followed by air-drying. The dried filters containing the suspensions were mounted on aluminium stubs using conductive carbon adhesive tape. *The S. polyrhiza* and *D. magna* specimens were processed similarly. All samples were rendered conductive by carbon coating before visualisation using SEM-EDX to investigate the interaction between PR-ENMs and aquatic organisms.

For transmission electron microscopy (TEM), samples were processed using conventional procedures and embedded in epoxy resin. Thin sections (~80 nm) were cut using a Leica uC7 ultramicrotome (Leica Microsystems, Vienna, Austria), collected on a Cu grid, and contrasted before observation with a JEOL JEM 1010 at 100 Kv or a JEOL-JEM 2100 HR-TEM at 200 Kv (JEOL, Tokyo, Japan) to visualise cellular integrity as well as potential internalisation of PR-ENMs.

### 3.6. Data Analysis

The EC/LC_50_ of the sample was determined using MicroBioTests Inc. (Gent, Belgium) software; in all cases, the EC/LC_50_ were determined at a 95% confidence limit. Statistical analysis and drawing of graphs were performed using GraphPad Prism8 version 8.4.3 for Windows (GraphPad Software, La Jolla, San Diego, CA, USA). A Student’s t-test and two-way ANOVA with a Tukey’s HSD post hoc test were applied to test the statistical difference between treatments at *α* = 0.05.

## 4. Conclusions

The toxicity investigation undertaken in this study illustrated that form-specific PR-ENMs and the respective ions both contributed to the adverse toxic effects observed in the biota at varying extents dependent on the type and form of the product and organism. Neither of the two forms (PR-ENMs or ions) were uniformly a principal toxicity agent. However, there was relatively good success per case in identifying the principal toxicity driver. The toxicity of PR-ENMs in descending order was as follows: SAN1 > SUN1 > SK1 > SUN2 > SUN3 > CA1, SUN1 ≥ SAN1 > SK1 > SUN3 > SUN2 > CA1, and SAN1 > SUN1 ≥ SK1 > SUN3 > SUN2 > CA1 for *P. subcapitata*, *S. polyrhiza*, and *D. magna*, respectively. For all organisms, SUN1 and SAN1 PR-ENMs and ions were more toxic, while the CA1 PR-ENMs and ions generally induced negligible effects.

Generally, *D. magna* and *S. polyrhiza* were the organisms with the highest and lowest sensitivities respectively to all PR-ENMs, except for CA1 and SAN1 PR-ENMs. *D. magna* was the only organism whose tissues were damaged by SUN1–3 PR-ENMs.

The binary PR-ENMs induced either synergistic or antagonistic effects, but the contribution of each of the ENMs was evident; hence we raise caution against the risk determination mixtures based on the EC/LC_50_ of a single ENM. The EC/LC_50_ determined for the respective test organisms varied, but were generally below 200 µg/L and 1000 µg/L for PR-ENMs and ions, respectively.

Risk characterisation indicated that no imminent toxicity risk was expected (RQ < 1) from PR-nAg (SAN1) and PR-nTiO_2_ (SUN2–SUN3), and Ag^+^ (SAN1, CA1, and SK1) and Zn^2+^ (SUN1). Instead, small to significant adverse effects in aquatic environments are expected. However, such findings serve as a valuable prioritisation guide, as they are from single dose studies, whereas repetitive and long-term exposure in actual environments may result in different scenarios. Considering that not all nanopollution emissions can be assessed for their environmental implications, such risk estimation can point to priority scenarios that require detailed assessments for a particular location.

Overall, the current study illustrates that, contrary to the low risk estimated by most in silico studies, the NEPs PR-ENMs and ions potentially pose small to potential ecological risks in the worst-case scenario or closer to the ENMs’ release and discharge source. While the current risk findings are not alarming, an increase in nanotechnology application will increase the exposure and consequently, the risk. Hence, mitigating measures that allow full nanotechnology exploitation and reduction of environmental and human health risks, such as safety by design, require more attention.

## Figures and Tables

**Figure 1 ijms-22-12468-f001:**
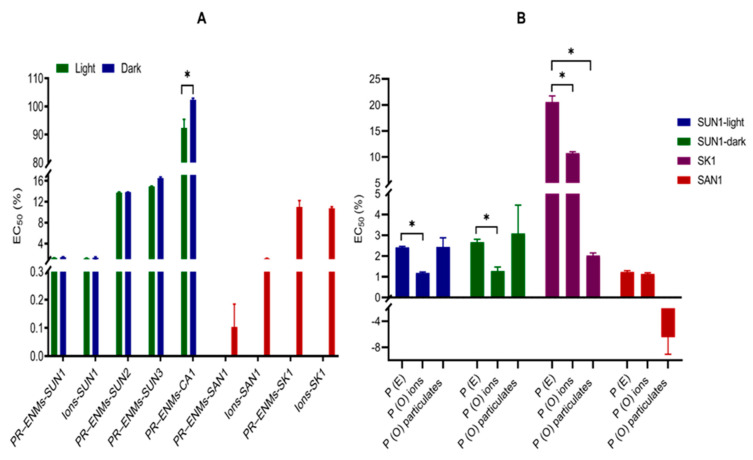
The 72 h EC_50_ values of *P. subcapitata* exposed to PR-ENMs and ions (**A**), and comparisons between the modelled effects (P (E)) and observed effects (P (O)) of particulates and ions on *P. subcapitata* (**B**). * indicates a significant difference at *p* ≤ 0.05. The significant difference in (**A**) was investigated between PR-ENMs obtained under light and dark conditions.

**Figure 2 ijms-22-12468-f002:**
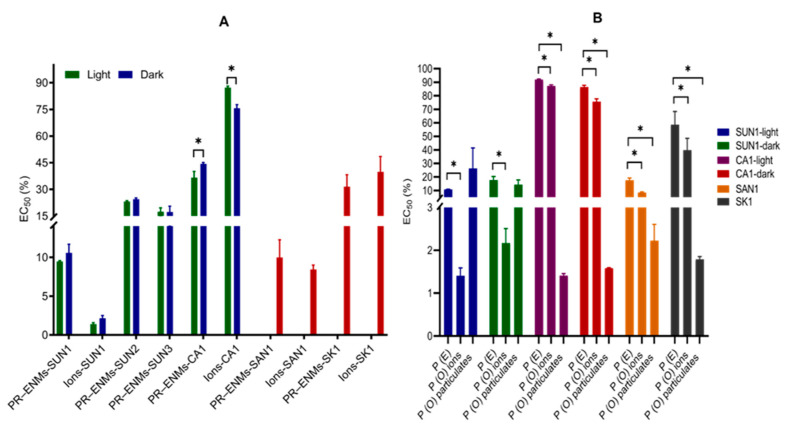
The 72 h EC_50_ values of *S. polyrhiza* after exposure to PR-ENMs and ions (**A**) and comparisons between the theoretical effects (*P* (E)) and observed effects (*P* (O)) of particulates and ions on *S. polyrhiza* (**B**). * indicates statistical difference at *p*-value of ≤0.05. The significant difference in image (**A**) was investigated between PR-ENMs obtained under light and dark conditions.

**Figure 3 ijms-22-12468-f003:**
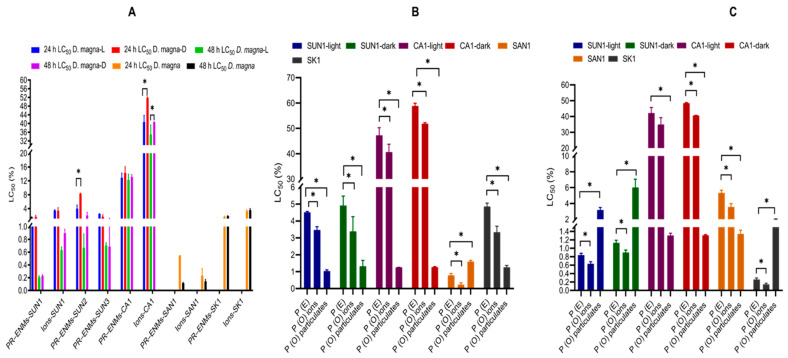
The 24 h and 48 h LC_50_ values of *D. magna* after exposure to PR-ENMs and ions (**A**) and comparisons of theoretical effects (P (E)) and observed effects (P (O)) of particulates and ions on *D. magna* at 24 h (**B**) and 48 h (**C**) exposure. L and D on the legends of image (**A**), respectively, denote PR-ENMs and ions obtained under light and dark conditions, while * indicates the statistical difference at *p*-value of ≤0.05.

**Figure 4 ijms-22-12468-f004:**
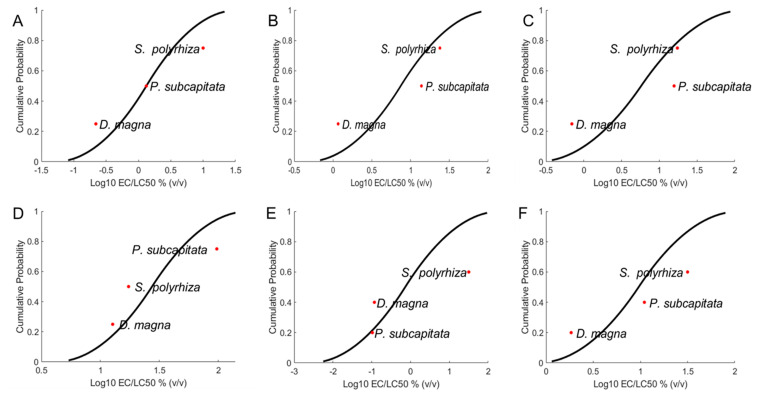
Species sensitivity distributions plot illustrating different responses of the three aquatic organisms to SUN1 (**A**), SUN2 (**B**), SUN3 (**C**), CA1 (**D**), SAN1 (**E**), and SK1 (**F**) PR-ENMs; *D. magna* and *S. polyrhiza* were the organisms with the highest and lowest sensitivity, except for CA1 (**D**) and SAN1 (**E**) PR-ENMs.

**Figure 5 ijms-22-12468-f005:**
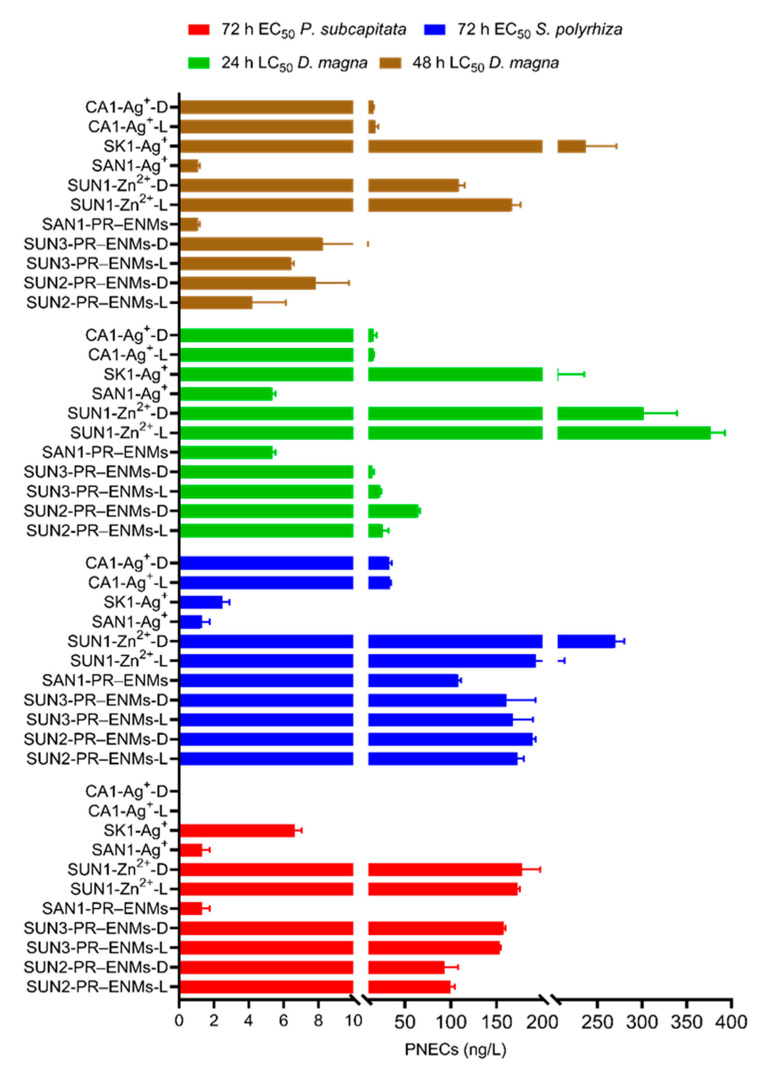
PNECs of PR-ENMs and ions were determined for *P. subcapitata*, *S. polyrhiza*, and *D. magna*. The L and D in the sample names (SUN2, SUN3, ions of SUN1, and CA1) denote PR-ENMs and ions obtained under light and dark conditions, respectively. The PNECs values plotted for SK1 for the test organisms *S. polyrhiza*, and *P. subcapitata* are in µg/L and not ng/L.

**Figure 6 ijms-22-12468-f006:**
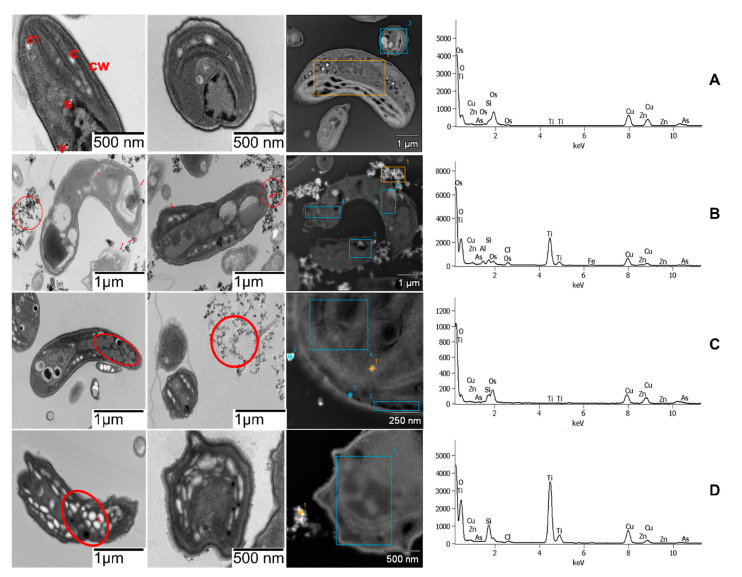
TEM-EDS images of cross-sections of unaffected *P. subcapitata* control (m = mitochondrion, c = chloroplast, s = starch grain, cw = cell wall, v = vacuole, n = nucleus) (row **A**) and *P. subcapitata* exposed to 10% (*v*/*v*) of PR-ENMs of SUN1 (row **B**), SUN2 (row **C**) and SUN3 (row **D**). The red arrows or circle indicate the adverse effects (slight morphological deformation, high lipid bodies, and starch grains) induced by SUNs PR-ENMs aggregates after 72 h exposure.

**Figure 7 ijms-22-12468-f007:**
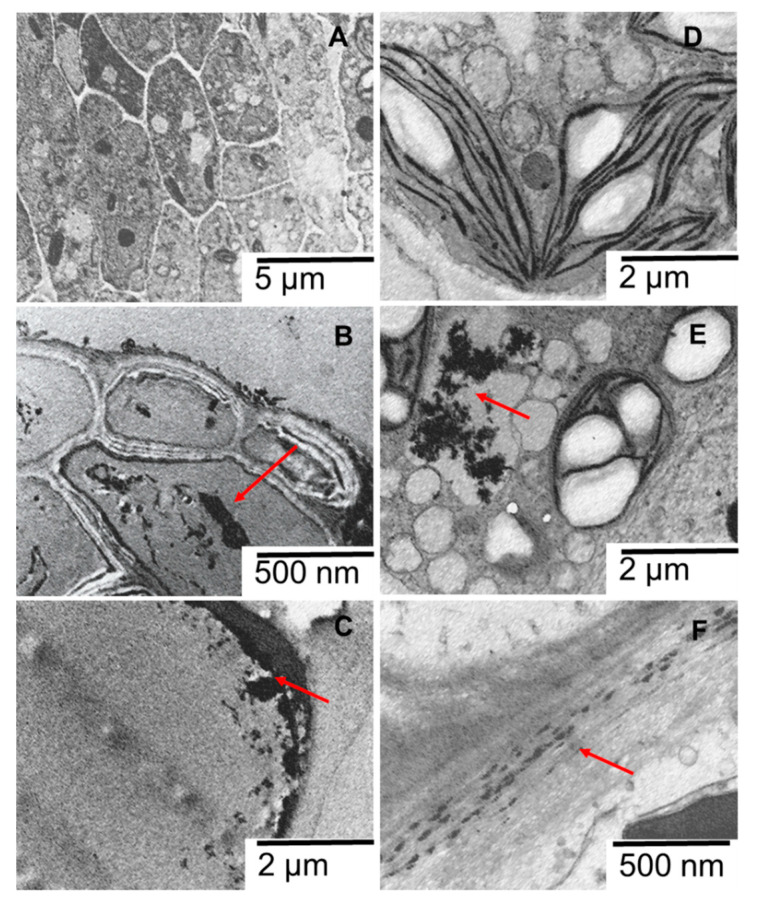
Cross-sectional images of the root (**A**) and frond (**F**) of control *S. polyrhiza*, and the root (**B**,**C**) and frond (**D**,**E**) of *S. polyrhiza* exposed to 100% (*v*/*v*) PR-ENMs of SUN1. The arrows indicate deposits suspected to be PR-ENMs.

**Figure 8 ijms-22-12468-f008:**
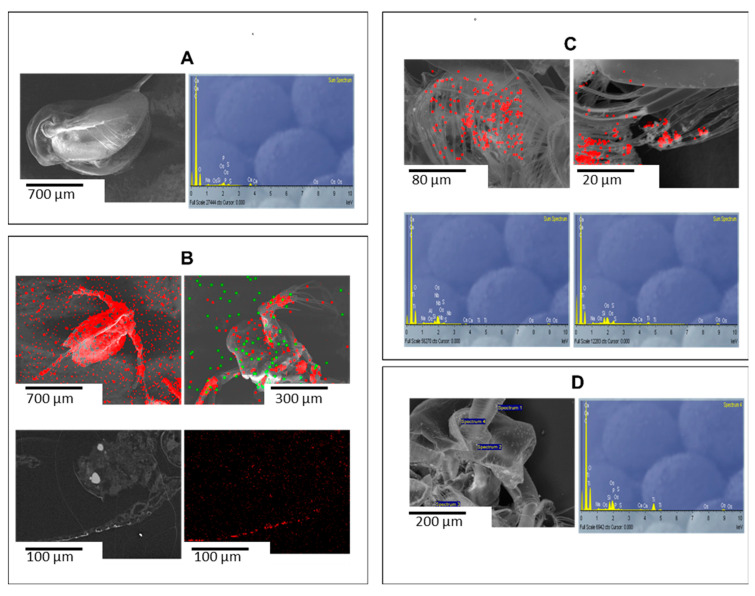
SEM-EDS images illustrating unaffected *D. magna* (**A**) control and adsorption of SUN1 (**B**), SUN2 (**C**) and SUN3 (**D**) PR-ENMs on morphological damaged *D. magna* after 48 h exposure (100% (*v*/*v*)).

**Table 1 ijms-22-12468-t001:** Determined EC/LC_50_ (µg/L) ± standard deviation of PR-ENMs and ions for *P. subcapitata*, *S. polyrhiza*, and *D. magna*.

Sample Name	Toxicant Type	72 h EC_50_		72 h EC_50_		24 h LC_50_		48 h LC_50_	
		*P. subcapitata*		*S. polyrhiza*		*D. magna*		*D. magna*	
		Light	Dark	Light	Dark	Light	Dark	Light	Dark
SUN1	Zn^2+^	173.1 ± 2.5	177.8 ± 20	239.1 ± 68	292.1 ± 40	376.4 ± 16	301.9 ± 37	167.0 ± 9.1	109.1 ± 6.0
SUN2	PR–nTiO_2_	99.50 ± 5.1	93.20 ± 15	173.0 ± 6.6	189.0 ± 3.4	26.03 ± 5.8	64.59 ± 2.1	4.205 ± 1.9	7.856 ± 1.9
SUN3	PR–nTiO_2_	153.5 ± 1.4	157.7 ± 2.2	167.5 ± 22	161.1 ± 31	23.18 ± 1.0	14.01 ± 2.4	6.453 ± 0.12	7.243 ± 0.57
CA1	Ag^+^	nd	nd	33.94 ± 1.2	32.25 ± 4.0	15.50 ± 0.7	15.80 ± 3.5	17.9 ± 3.0	15.89 ± 0.36
SK1	Ag^+^	663.5 ± 39		2494 ± 397		206.2 ± 29		236.9 ± 34	
SAN1	PR–nAg	1.315 ± 0.45		108.2 ± 3.1		5.367 ± 0.17		1.110 ± 0.08	
	Ag^+^	9.387 ± 0.26		71.74 ± 5.0		2.089 ± 0.65		1.065 ± 0.31	

nd = not determined.

**Table 2 ijms-22-12468-t002:** The physicochemical properties of the PR-ENMs and ions examined for toxic effects as reported in [15]. The particle size is reported in terms of (width × length).

Sample Name	ENMs Type	Target Analyte	Concentration (mg/L)		ENMs Shape	ENMs Size (nm)
			Light Condition	Dark Condition		
SUN1	nTiO_2_ + nZnO	Ti	6.99 ± 0.06	8.51 ± 0.209	elongated	13 × 69
		Zn	26.8 ± 0.39	27.00 ± 0.84	angular	34 × 30
		Zn^2+^	13.7 ± 0.42	13.84 ± 0.55		
SUN2	nTiO_2_	Ti	0.68 ± 0.03	0.78 ± 0.02	angular	40 × 21
SUN3	nTiO_2_	Ti	0.96 ± 0.02	0.93 ± 0.009	angular	30 × 21
CA1 *	nTiO_2_ + nAg	Ti	39.52 ± 0.82	30.71 ± 0.71	angular + elongated	8 × 34, 17 × 93
		Ag	56.19 ± 3.0	52.38 ± 0.8	angular	22 × 20
		Ag^+^	39.05 ± 1.6	33.33 ± 6.4		
SAN1	nAg	Ag	0.95 ± 0.03		spherical	23 × 20, 9 × 10
		Ag^+^	0.82 ± 0.01			
SK1	nTiO_2_ + nAg	Ti	5.73 ± 0.23		angular	62 × 51
		Ag	6.00 ± 0.4		spherical + angular	8 × 8 nm, 89 × 95
		Ag^+^	6.40 ± 0.4			

* indicates that the concentrations of the CA1 analytes are reported in (µg/L).

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
