# Peer review of "Aquatic Toxicity Effects and Risk Assessment of ‘Form Specific’ Product-Released Engineered Nanomaterials"

_ijms, 2021, doi:10.3390/ijms222212468_

Round 1
Reviewer 1 Report
In the manuscript “Aquatic Toxicity Effects and Risk Assessment of ‘Form Specific’ Engineered Nanomaterials Released from Nano-Enabled Products” the authors aimed to investigate the toxicity effects and risk of form specific PR-ENMs and associated ions. In addition, they assessed the interactions between sunscreen PR-ENMs and aquatic biota (Pseudokirchneriella subcapitata, Spirodela polyrhiza, and Daphnia magna) and examined some ultrastructural abnormalities resulting from exposure.
The topic is very interesting and this manuscript could provide important methodological and scientific information that are of wide interest.
The manuscript is well written and the experimental design appears very clear. I well appreciated all the figures, that are of high quality.
Author Response
We have provided improved descriptions of the Figures which has improved the quality of the information of the figures. Additionally, the manuscript has undergone an extensive round of editing.
Reviewer 2 Report
The work by Lehutso et al discusses the toxic effects of engineered nanomaterials on aquatic species. The topic is of high interest with the increase of
Engineered nanomaterials (ENMs) can be released throughout the life cycle of nano-enabled products (NEPs) into the environment, and water resources are among the most common recipients together with their habitat. Nanotechnologies are developing fast, enter in all fields of our daily life and the Biosphere is affected in an unprecedented way. All efforts toward investigation and evaluation of nanoparticles pollution are of utmost importance. Therefore, the work deserves its place in IJMS, MDPI.
Advise the authors to follow the below-mentioned recommendations to improve the quality of their work and thus deserve its place in the journal.
The Introduction is informative and easy to read. The Results and Discussion part needs though fine edit. I advise the authors to start this part with a more descriptive paragraph on their work and its meaning for the field. In the current version, this part starts with a telegraphic beginning, which devoid the work of descriptive meaning.
The species names should be in Italic.
I advise the authors to try to present and discuss the results concerning different species toxicity effects together without subheadings that divide the aquatic species. It is very important to provide a short statement of why these aquatic species have been used.
The Materials and Methods section: I advise the authors to replace the first part of 3.1. with the beginning of Results and Discussion. The M and M section has to have only well-described experimental procedures. In this version of the manuscript, this section sounds like a Results’ discussion.
The same is relevant for Table 2: it has to be placed in the Results section together with all descriptive text regarding its content.
Line 452: Please, give a reference for this statement. Which are these preliminary data on which you plan these experiments?
I advise the authors to explain why for the different species they have used different time intervals for studying the toxicity effect of the tested nanomaterials.
This is important to be discussed in The M and M section as well as in the Results section that deals with these explanations.
Figure 4 needs a more descriptive Figure caption.
Figure 6 has no A, B, C etc as mentioned in the text.
Moreover, it requires a more descriptive caption. Furthermore, the red signs are good for indicating the alterations and accumulation of the studied ENMs and are not suitable for the figure size bars.
The same is valid for all the following microscopy images. The red colours for scale bars are highly unsuitable.
There are typos and stylistic errors. I advise the authors to proofread the text.
Author Response
Reviewer 2 comments |
Authors response and action |
The Results and Discussion part needs though fine edit. I advise the authors to start this part with a more descriptive paragraph on their work and its meaning for the field. In the current version, this part starts with a telegraphic beginning, which devoid the work of descriptive meaning |
Accepted, opening paragraph provided to describe the Results and Discussion section, Line 132-142. However, the authors do not support the suggestion of providing the summary of the meaning of the work before the results are presented and discussed. |
The species names should be in Italic |
Accepted. Species names are written in italics throughout the manuscript. |
I advise the authors to try to present and discuss the results concerning different species toxicity effects together without subheadings that divide the aquatic species. It is very important to provide a short statement of why these aquatic species have been used |
Partially accepted: justification for the use of the organisms has been given in lines 114-117.
However, the authors do not agree with the suggestion of bundling the results; that is more a preference and bears no implication on the integrity of the results. Furthermore, the authors hold the view that the separation per species aids navigate the reader to specific results and discussion, thus reducing confusion potential. We explored the alternative option of reporting and discussing the results was by NEPs type, this route was deemed not viable due to the many subheadings (6 NEPs heading vs. 3 organism type) and repetition in discussion.
Similar to Section 2.1, Section 2.3, the results and discussion of PR-ENMs and sunscreen interaction are discussed per organism group, because it allows easy comparison between the exposed organisms and control. Alternative reporting will result in the repetition of control images.
|
I advise the authors to replace the first part of 3.1. with the beginning of Results and Discussion. The M and M section has to have only well-described experimental procedures. In this version of the manuscript, this section sounds like a Results’ discussion. |
Partially accepted: The beginning of the previous manuscript version was the synopsis of the Results and Discussion, thus not suitable to be moved to Materials and Methods. Instead, a new sentence is added in the M and M is added, lines 469-444. The methods used herein were previously published and, to avoid plagiarism and repetition, the methods, and the physicochemical properties of the PR-ENMs and ions were briefly described, and sources cited. Lastly, the entire M and M section was thoroughly edited. |
The same is relevant for Table 2: it has to be placed in the Results section together with all descriptive text regarding its content. |
Table 2 and associated content appear in the Methods and Materials section because they are not the findings in the current paper. Rather, these are the results which are already published elsewhere (refs 114 and 15), but they are the starting material of the current paper objectives. To provide more clarity about the source of the table, the refs are now included in the Table 2 caption, lines 480. |
Line 452: Please, give a reference for this statement. Which are these preliminary data on which you plan these experiments? |
Accepted by providing clarity: The preliminary data was not sourced from the literature. Instead, we empirically investigated the exposure concentration range (2-fold dilution). To avoid confusion caused, the sentence was edited to show that the preliminary data were empirically investigated, lines 508-509. |
I advise the authors to explain why for the different species they have used different time intervals for studying the toxicity effect of the tested nanomaterials. This is important to be discussed in The M and M section as well as in the Results section that deals with these explanations. |
Advice accepted and discussed in the introduction section, lines 114-117 and R&D, line 134-138. To avoid repetition, the reasons were not presented in M&M. |
Figure 4 needs a more descriptive Figure caption. |
Accepted: the caption of Figure 4 is changed to be more descriptive, line 236-238. |
Figure 6 has no A, B, C etc as mentioned in the text. Moreover, it requires a more descriptive caption. Furthermore, the red signs are good for indicating the alterations and accumulation of the studied ENMs and are not suitable for the figure size bars. |
Accepted: The A, B, C, and D in Figure 6 are added and the red color on the scale bars is changed, and the caption of Figure 6 is changed to be more descriptive lines 405-408. |
The same is valid for all the following microscopy images. The red colours for scale bars are highly unsuitable. |
Accepted: the red color on the scale bars is removed on all microscopy images, Figure 7 (line 433), Figure 8 (line 458) and Figure S3 (supplementary material line 37). |
There are typos and stylistic errors. I advise the authors to proofread the text. |
The entire manuscript was proofread, and typographical and stylistic errors corrected. |
Round 2
Reviewer 2 Report
I agree with the authors' comments on the review.
A minor suggestion is to include in Figures 6, 7 and 8 a black line indicating the scale. At the present form, it seems like the text box in white serves as a scale bar.
In a summary, I suggest acceptance of the manuscript.
Author Response
Reviewer comments |
Authors response and action |
A minor suggestion is to include in Figures 6, 7 and 8 a black line indicating the scale. At the present form, it seems like the text box in white serves as a scale bar. |
Comment accepted. Scale bar line was added on Figures 6 (line 405), Figure 7 (line 434), Figure 8 (line 458) and Figure S3 Line 36. |